# Towards Faithful and Controllable Personalization via Critique-Post-Edit Reinforcement Learning

## Abstract

Faithfully personalizing large language models (LLMs) to align with individual user preferences is a critical but challenging task. While supervised fine-tuning (SFT) quickly reaches a performance plateau, standard reinforcement learning from human feedback (RLHF) also struggles with the nuances of personalization. Scalar-based reward models are prone to reward hacking which leads to verbose and superficially personalized responses. To address these limitations, we propose **Critique-Post-Edit**, a robust reinforcement learning framework that enables more faithful and controllable personalization. Our framework integrates two key components: (1) a **Personalized Generative Reward Model (GRM)** that provides multi-dimensional scores and textual critiques to resist reward hacking, and (2) a **Critique-Post-Edit** mechanism where the policy model revises its own outputs based on these critiques for more targeted and efficient learning. Under a rigorous length-controlled evaluation, our method substantially outperforms standard PPO on personalization benchmarks. Personalized Qwen2.5-7B achieves an average 11% win-rate improvement, and personalized Qwen2.5-14B model surpasses the performance of GPT-4.1. These results demonstrate a practical path to faithful, efficient, and controllable personalization.

## 1 Introduction

### 1.1 Personlization

As large language models (LLMs) evolve from general-purpose assistants to personalized agents, the ability to tailor responses to a user's unique attributes, needs, and constraints has become a critical frontier (Tan et al., 2025; Zhao et al., 2025).Nonetheless, prevailing paradigms are largely limited to (i) post-training that align LLMs with universal values and preferences (Ouyang et al., 2022), or (ii) rigid retrieval over a personal knowledge base which is then superficially incorporated in the responses. Both tend to feel unnatural and brittle—either enforcing one-size-fits-all persona or sprinkling persona factoids as references—without a deeper, integrated understanding of the user. True personalization requires meta understanding: the model must not only learn what to emphasize or omit, but also adapt to each individual's unique persona. This profound understanding must then be expressed through nuanced changes in wording, structure, and detail.

However, current optimizing methods fall short for cultivating such meta understanding. Supervised fine-tuning (SFT) (Raffel et al., 2020) and direct preference optimization (DPO) (Rafailov et al., 2023) provide limited supervision: models quickly saturate on available labels and still struggle to internalize "what counts" as personalization beyond verbatim reference, keywords or templates. Policy-gradient based Reinforcement Learning (RL) (Ouyang et al., 2022) like PPO (Schulman et al., 2017) and GRPO (Shao et al., 2024) with outcome/value-based rewards also struggles since rewards are sparse and prone to be hacked (Wang et al., 2023) (Sun et al., 2023). In practice, standard Bradley-Terry(BT) based reward models frequently incentivize undesirable behaviors—such as verbose outputs and generic stock phrases (Sun et al., 2025) (Bu et al., 2025), leading to reward hacking rather than faithful personalization.

On the other hand, Generative Reward Model (GRM) (Zhang et al., 2024b) changes this landscape. Instead of a single scalar, a GRM could produce rigorous rationale along with multi-dimensional

scores that explain what to improve and why. As a verifier, the GRM's textual judgments provide a more robust and nuanced feedback signal, substantially reducing susceptibility to reward hacking compared to Bradley-Terry based reward models. Recent tool-integrated RL post-training works has bugun exploring the potential of GRMs on deep-research, web-browsing, and code-execution tasks (Xu et al., 2025; Wu et al., 2025b). However, to our knowledge, GRM has not yet been systematically explored for personalization, where reasoning about user-specific preferences is central. The combination of detailed critiques and multi-facet reward is well-suited for personalization as it provides a more subtler and instructive supervisory signal.

Inspired by Helpsteer3 (Wang et al., 2025), we introduce a critique-post-edit RL paradigm that leverage the critiques from a personalized GRM. The policy model first generate an initial response based on the given query and persona traits; the GRM evaluates the response and produces a critique according to the query and persona; the policy refines its original response by incorporating the feedback from the critique. We compute rewards for both original and edited responses and update the policy with a batch that mixes on-policy and edited (off-policy) samples. This yields two benefits. First, the learning signal becomes diverse and targeted: advantages are estimated over multiple, concrete improvement paths rather than a single outcome, improving training stability. Second, it matches the nature of personalization: there is no single golden answer for a given query and its corresponding user. Multiple nuanced, equally valid ways can reflect the user's preferences. Critique-Post-Edit RL explicitly exposes those alternatives during training, helping the policy acquire subtle, faithful personalization.

We conduct a comprehensive evaluation of our PersonalizedLLMs on the PersonaFeedback (Tao et al., 2025), AlpacaEval (Dubois et al., 2024), and PersonaMem (Jiang et al., 2025) benchmarks. With a rigorous length-controlled evaluation protocol (Dubois et al., 2024) to mitigate scoring biases, our approach demonstrates significant gains. Our Qwen2.5-7B model achieves an average win-rate improvement of 14% over a strong PPO baseline. Moreover, our Personalized-Qwen2.5-14B model not only matches this improvement but also surpasses the performance of GPT-4.1, highlighting the effectiveness and scalability of our framework for building faithfully personalized models.

Our contributions are as follows:

- We identify the limitations of SFT/DPO and Bradley-Terry based RMs for personalization.

- We train a personalized GRM that provides multi-dimensional scores and actionable critiques that achieved SoTA results on the PersonaFeedback Benchmark.

- We introduce a **Critique-Post-Edit** RL framework that leverages GRM feedback[1] to refine responses and learn from a group of diverse mixed on-policy(original) and off-policy(refined) responses, yielding more nuanced and faithful personalization.

- Under rigorous length-controlled evaluation, our approach delivers strong gains over PPO and surpasses GPT-4.1, demonstrating a practical path to controllable and scalable personalization.

## 2 RELATED WORK

Personalization in LLMs aims to tailor responses to individual users' profiles, preferences, and contexts, thereby improving user satisfaction and engagement. Early approaches focused on persona-conditioned dialogue generation (Zhang et al., 2018; Song et al., 2019), where predefined user traits are used to guide response generation. Meta-learning has also been explored as a way to enable models to quickly adapt to new users with limited supervision (Madotto et al., 2019). More recent studies investigate richer modeling of user information, for example by combining sparse and dense persona representations (Tang et al., 2023).

Complementary to persona conditioning, retrieval-augmented approaches personalize by fetching information from user knowledge base and injecting it into prompts. Salemi et al. (2024) propose such a RAG framework and further extend it by supervised tuning the LLM with feedback loop (Salemi & Zamani, 2025).

---

[1]For clarity, we use the terms "feedback" and "critique" interchangeably in this work.

Benchmark efforts such as PersonaBench (Tan et al., 2025) evaluate models on their ability to handle synthetic private user data, emphasizing the importance of faithful and controllable personalization.

Additionally, in other domains, such as education and enterprise applications, personalized assistants have also been widely studied (Li et al., 2023; Mysore et al., 2023; Lu et al., 2024; Zhang et al., 2024a).

Despite growing interest, personalization remains relatively underexplored compared to general alignment. In particular, reinforcement learning pipelines often lack the granularity required to capture user-specific nuances, with only recent attempts exploring dynamic profile modeling for personalized alignment (Zhao et al., 2025).

## 3 PIVOTAL STUDY

### 3.1 PRIMARY ATTEMPTS

We use approximately 18k annotated samples as the primary dataset. SFT, DPO, and RL all adopt a batch size of 128, with SFT and DPO trained for 3 epochs and RL for 2 epochs. SFT and DPO use around 54k (18k × 3) training samples, while RL generates around 32k (18k × 2) trajectories.

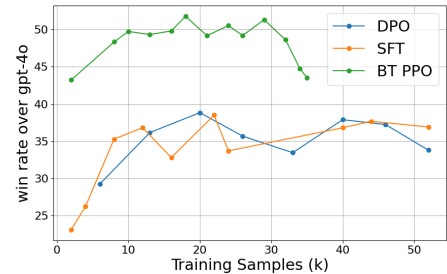

As illustrated in Figure 1, we observed that SFT and DPO performance[2] quickly plateaus with increasing data. On the other hand, PPO with Bradley-Terry RM continues to yield substantial gains.

Figure 1: Performance curve of SFT, DPO and RL with increasing data size.

However, we also observe that BT-guided PPO encounters severe **reward hacking**, where reward model, even the judge LLM, are gamed by superficial clues rather than genuine improvements.

As shown in Figure 2, policy model tends to add a short notice after answering which yields a significant increase of reward score despite limited improvement.

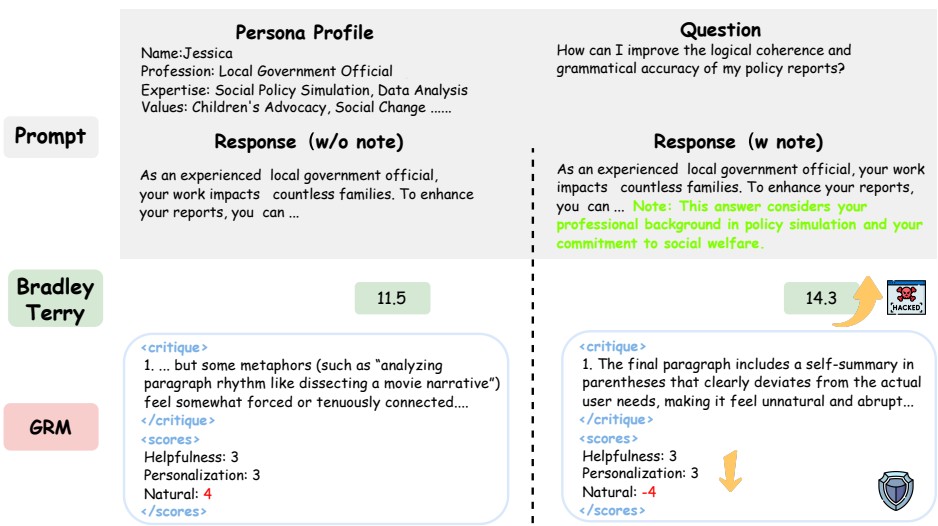

Figure 2: An illustrative "reward hacking" case from RL training with a BT reward model. The model learns to exploit a shortcut by explicitly mention persona traits to get higher reward scores.

---

[2]Performance means length controled win rate over gpt-4o, details about this metric are provided in Section 5.1

## 3.2 ANALYSIS

To tackle with reward hacking, we adopt a Generative Reward Model (GRM) that must first produce a concise critique before giving the final scores. This textual verification makes the reward score more robust in "seemingly personalized" cases, where superficial cues would otherwise be rewarded by Bradley-Terry based RM.

Upon further analysis, we find that the gap between a less-personalized and a truly personalized response is often subtle and may only require small, targeted modifications. We take inspiration from HelpSteer3 (Wang et al., 2025), which use editing and feedback pipeline to improve model's performance on open-ended question. These observations motivates a Critique-Post-Edit training paradigm: the GRM provides actionable critiques; the policy edits its own output accordingly; and training leverages both the original and edited responses.

## 4 METHOD

Building on the motivation discussed in Section 3, we introduce a Critique-Post-Edit framework where the GRM provides both scalar rewards and textual feedback, enabling the policy model to generate improved edited responses based on this feedback, as illustrated in Figure 3.

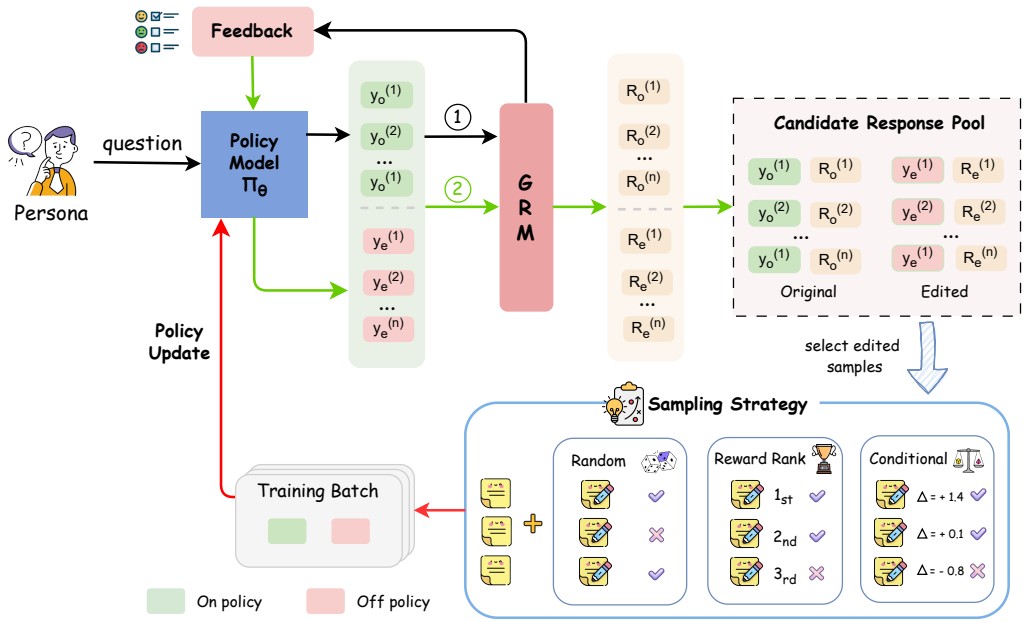

Figure 3: Overview of the Critique-Post-Edit framework. (1) The policy model $\pi_\theta$ generates an original response $y_o$. The GRM evaluates $y_o$ and provides critique, guiding the model to create an edited response $y_e$. (2) The GRM then provides rewards for both $y_o$ and $y_e$, denoted as $R_o$ and $R_e$, forming a candidate pool from which a sampling strategy selects edited samples to combine with the original ones into a training batch for the policy update. Sampling strategies include: Random (selecting a random subset), Reward Rank (selecting the highest-reward samples), and Conditional (selecting if the reward exceeds the original reward).

### 4.1 DETAILS ABOUT GRM

To train the GRM, we build upon the preference data described in Section 5.1 by conducting a second round of annotation. For each response in the preference pairs, we use GPT-4o-mini to provide detailed critique, which includes: (1) actionable suggestions for improvement (critiques), and (2) scores for three distinct dimensions—helpfulness, personalization, and naturalness—on a scale from -5 to +5.

The GRM takes a tuple consisting of a `(question, user profile, response)` as input and is trained on this dataset to produce a twofold output: (1) a natural language critique with specific improvement suggestions, and (2) a set of scores for the following three dimensions:

- **Helpfulness**: Assesses whether the response is accurate, comprehensive, and effectively solves the user's problem, with practical and in-depth content.
- **Personalization**: Evaluates the appropriate use of user information, avoiding mechanical stacking or irrelevant details. The content should align with the user's background and needs.
- **Naturalness**: Judges whether the writing style is fluent and comfortable, with a natural tone that matches daily communication, avoiding robotic or verbose expressions.

To create a unified quality metric, we calculate a final weighted score $S_{\text{final}}$ using the formula:

$$S_{\text{final}} = w_h \cdot S_{\text{helpfulness}} + w_p \cdot S_{\text{personalization}} + w_n \cdot S_{\text{naturalness}}$$

where $w_h, w_p$, and $w_n$ represent the weights for each respective dimension. Finally, we filter out any data instances where the final weighted scores are identical, thus constructing the final GRM training dataset of 17K examples.

## 4.2 Details about Feedback edit

We introduce a **Critique-Post-Edit Framework**. As illustrated in Figure 3, for a given input query $q$, the policy model $\pi_\theta$ first generates a set of candidate responses $\{y^{(1)}, ..., y^{(k)}\}$ through multiple rollouts. These responses are then evaluated by the GRM, which produces a scalar reward $R^{(i)}$ and textual feedback $f^{(i)}$ for each response. This feedback is concatenated with the original query and its corresponding response to form a new prompt, which is feedback to the policy model to encourage it to revise its output based on the feedback provided, resulting in an edited response $y_{\text{e}}$. This process ultimately builds a sample pool that contains original and edited responses for policy updates.

### 4.2.1 Sampling Strategy

We construct PPO training batches from a sample pool that includes both the original responses $\{y_{\text{o}}\}$ and the edited responses $\{y_{\text{e}}\}$. To maintain policy stability, all sampling strategies retain the full set of original responses. We explore the following three sampling methods:

- **Random Sampling**: Randomly selects a subset of edited responses $\{y_{\text{e}}\}$ to be included in the training batch, according to a predefined sampling rate $r_{\text{e}}$.
- **Reward Rank Sampling**: For each query, The edited responses are sorted in descending order based on their reward scores. The top $r_{\text{e}}$-proportion of these responses are selected to form a high-quality candidate pool.
- **Conditional Sampling**: For each query, the edited responses are sorted in descending order based on their reward scores or the improvement margin over the original response. The top $r_{\text{e}}$-proportion of responses in terms of reward increase are included in the candidate pool.

### 4.2.2 Hybrid Policy Update Loss

Since the training batch consists of both on-policy original responses and off-policy edited responses, applying the standard PPO loss directly may lead to instability due to distributional mismatch. To address this, we design a **hybrid policy update loss** that treats these two types of data differently.

For each sample $y$ in the batch, the policy gradient loss $\mathcal{L}_{PG}(y)$ is defined as:

$$\mathcal{L}_{PG}(y) = \begin{cases} -\min\left(r_t(\theta)\hat{A}_t, \text{clip}(r_t(\theta), 1-\epsilon, 1+\epsilon)\hat{A}_t\right) & \text{if } y \in \mathcal{D}_{\text{o}} \\ -\text{clip}\left(\frac{\pi_\theta(y)}{\pi_{\text{e}}(y)}, 1-\epsilon_{\text{low}}, 1+\epsilon_{\text{high}}\right)\hat{A}_t & \text{if } y \in \mathcal{D}_{\text{e}} \end{cases}$$

Here, $\mathcal{D}_{\mathrm{o}}$ and $\mathcal{D}_{\mathrm{e}}$ denote the sets of original and edited samples, respectively, and $\hat{A}_t$ is the estimated advantage.

Specifically, for samples in $\mathcal{D}_{\mathrm{o}}$, we apply the standard PPO-Clip loss, which stabilizes learning by clipping the importance weight $r_t(\theta) = \frac{\pi_\theta(y)}{\pi_{\theta_{\mathrm{old}}}(y)}$ to stay within a trust region.

For samples in $\mathcal{D}_{\mathrm{e}}$, we treat them as off-policy data. Their importance weights are corrected using the ratio between the current policy $\pi_\theta$ and the implicit editing policy $\pi_{\mathrm{e}}$ that generated these samples. To mitigate the potential training instability arising from the high-importance weights, this ratio is clipped to stay within a trust region of $[1-\epsilon_{\mathrm{low}}, 1+\epsilon_{\mathrm{high}}]$. Logarithmic probabilities $\log \pi_{\mathrm{e}}(y)$ are pre-computed and retained during the editing stage, allowing for a principled correction for off-policy training.

## 5 EXPERIMENTS

### 5.1 IMPLEMENTATION DETAILS

**Training Datesets** We first create a pool of questions comprising 10K specific questions and 10K general questions, following the methodology of PersonaFeedback (Tao et al., 2025). For each question, we prompt 5 different LLMs to generate a variety of responses. We then use GPT-4o as a judge to assign a holistic quality score on a scale of 1 to 5 for each response, retaining only pairs where the absolute score difference exceeds a threshold of 2, which ensures that our training data contain preference pairs with clear, yet varied quality gaps. In the SFT stage, we use the chosen responses $y_c$ as target labels.Please notice that, in SFT stage, we did not train our model how to refine its answer. In the RL stage, we train the initialized SFT model using PPO.

**Model** All models in our experiments, including policy models, GRMs, and BT reward models, are based on the Qwen2.5-Instruct series (Qwen et al., 2025). Specifically, we conducted comprehensive experiments and analysis to obtain personalized GRMs. By scaling up model parameters, our personalized GRM achieved SoTA results on the PersonaFeedback Benchmark comparing to BT models and proprietary LLMs. Empirical results and analysis could be found in the Appendix B. For policy model, we trained our personalized LLMs based on Qwen2.5-7B and Qwen2.5-14B model with a 14B personalized GRM. Further details on specific hyperparameters are provided in the AppendixA.

**Evaluation** For evaluation, we sample a subset from the PersonaFeedback Benchmark rather than using the complete dataset. Specifically, we randomly select 50 questions from each difficulty tier (easy, medium, hard) for both Specific and General categories, resulting in a total of 300 evaluation samples.

To provide a comprehensive evaluation, we also test our models on Alpaca and PersonaMem benchmarks. For the Alpaca benchmark, we construct a persona for each question in alpaca eval that would realistically ask such a question, using the alpaca eval question and persona as the input. For the PersonaMem benchmark, we use the provided persona attributes as the persona information, which is then combined with the question as the input for the model. Throughout all experiments, we use GPT-4o as the single fixed baseline.

To address the well-known length bias in LLM-as-judge protocols, we adopt the length-controlled evaluation framework from Dubois et al. (2024). This method uses a Generalized Linear Model (GLM) to explicitly remove length as a confounding factor via regression-based debiasing.

Following official recommendations [3], we employ the `length_controlled_minimal` variant. The probability that model $m$ wins against baseline $b$ on input $x$ is modeled as:

$$q_{\min}(y = m \mid z_m, z_b, m, b, x) := \text{logistic} \left( \theta_m - \theta_b + \phi_{m,b} \cdot \tanh \left( \frac{\text{len}(z_m) - \text{len}(z_b)}{\text{std}(\text{len}(z_m) - \text{len}(z_b))} \right) \right)$$

---

[3]https://github.com/tatsu-lab/alpaca_eval/issues/346

where $\theta_m - \theta_b$ captures the inherent ability difference between models, and $\phi_{m,b}$ parameterizes the length effect. By setting the length effect term to zero during evaluation, we obtain length-controlled win rates that provide fairer comparisons. Detailed mathematical formulations are provided in Appendix D.

## 5.2 MAIN RESULTS

Table 1: Comparison of Open Source Models with Proprietary Models on PersonaFeedback, Alpaca, and PersonaMem Benchmarks. Our Critique-Post-Edit framework uses 0.5 sampling ratio with Random Sampling strategy.

| Model | PersonaFeedback | | | | | | | | | Alpaca | PERSONAMEM |
| | Specific | | | | General | | | | Total Avg | | |
| | Easy | Mid | Hard | Avg | Easy | Mid | Hard | Avg | | | |
| --- | --- | --- | --- | --- | --- | --- | --- | --- | --- | --- | --- |
| *Proprietary Models* | | | | | | | | | | | |
| GPT-4.1 | 66.4 | 60.9 | 55.8 | 61.0 | 66.7 | 64.4 | 60.8 | 64.0 | 62.5 | 64.5 | 49.5 |
| GPT-4o-mini | 51.1 | 52.9 | 49.1 | 51.0 | 49.8 | 47.2 | 37.9 | 45.0 | 48.0 | 49.7 | 38.2 |
| Qwen3-8B | 50.3 | 44.2 | 32.0 | 42.2 | 34.4 | 33.1 | 30.5 | 32.7 | 37.5 | 42.4 | 25.8 |
| Qwen3-14B | 57.6 | 51.2 | 46.2 | 51.7 | 48.5 | 45.8 | 36.9 | 43.7 | 47.7 | 47.3 | 32.7 |
| Qwen3-32B | 61.0 | 53.3 | 51.5 | 55.3 | 56.6 | 48.1 | 40.1 | 48.3 | 51.8 | 53.1 | 42.7 |
| Qwen2.5-32B | 56.0 | 52.6 | 39.0 | 49.2 | 50.8 | 42.6 | 36.8 | 43.4 | 46.3 | 48.7 | 36.5 |
| *Open Source Models* | | | | | | | | | | | |
| Qwen2.5-7B | 36.7 | 33.9 | 30.2 | 33.6 | 31.9 | 29.4 | 25.5 | 28.9 | 31.2 | 31.0 | 20.2 |
| +SFT | 42.9 | 34.5 | 34.0 | 37.1 | 46.1 | 37.8 | 35.8 | 39.9 | 38.5 | 35.0 | 26.2 |
| +DPO | 40.6 | 38.1 | 34.8 | 37.8 | 45.6 | 34.4 | 32.6 | 37.5 | 37.6 | 34.2 | 30.6 |
| +PPO | 58.0 | 51.4 | 49.1 | 52.8 | 58.7 | 52.9 | 48.9 | 53.5 | 53.1 | 40.6 | 33.6 |
| Ours | 73.2 | 69.7 | 64.7 | 69.2 | 61.2 | 59.7 | 56.2 | 59.0 | 64.1 | 54.5 | 50.2 |
| Qwen2.5-14B | 42.3 | 31.3 | 30.9 | 34.8 | 37.0 | 34.0 | 28.6 | 33.2 | 34.0 | 44.1 | 23.1 |
| +SFT | 45.5 | 38.4 | 34.2 | 39.4 | 53.1 | 45.1 | 44.8 | 47.7 | 43.5 | 47.6 | 31.4 |
| +DPO | 49.6 | 39.8 | 37.9 | 42.4 | 48.1 | 40.0 | 38.6 | 44.3 | 42.3 | 48.5 | 30.8 |
| +PPO | 74.9 | 66.7 | 54.5 | 65.4 | 68.9 | 65.8 | 60.1 | 57.8 | 61.6 | 53.8 | 44.6 |
| Ours | **80.9** | **79.4** | **71.8** | **77.4** | **79.9** | **74.6** | **73.9** | **76.1** | **76.8** | **64.0** | **67.1** |

Table 1 shows that our Critique-Post-Edit framework brings substantial improvements over standard PPO training. The 7B model sees a jump from 53.5% to 64.1% in the length-controlled win rate, a gain of more than 11 points. The 14B model performs even better, climbing from 65.2% to 76.8%. Similar consistent gains are observed on the Alpaca and PersonaMem benchmarks, demonstrating the robustness of our approach in diverse personalization scenarios. In particular, our 7B model beats GPT-4o-mini by a wide margin (64.1% vs 48.0%), while the 14B version clearly outperforms GPT-4.1. These gains hold consistently across both specific and general questions, suggesting that our approach works well in different personalization settings.

To validate our evaluation methodology, we recruited three human experts to independently evaluate samples from PersonaFeedback and selected GPT-4.1 as our primary evaluation baseline due to its high consistency with human evaluators, details see Appendix C. We computed Cohen's Kappa to measure interrater agreement across three comparisons: (1) **Model-Human**: GPT-4.1 vs human experts achieved an average $\kappa = 0.71$; (2) **Model-Model**: GPT-4.1 vs other models (GPT-4o-mini, GPT-4.1-mini, DeepSeek-v3, Claude-3.5-Sonnet) averaged $\kappa = 0.67$; (3) **Human-Human**: The three experts achieved an average $\kappa = 0.70$. These substantial agreement levels validate the reliability of our evaluation methodology.

## 5.3 ABLATION

Table 2: Ablation on Reward and Feedback Model: Impact on Performance.

| Setting | Length-controlled Win Rate | Win Rate | Response Length |
| --- | --- | --- | --- |
| BT wo/edit | 51.78 | 51.65 | 995 |
| GRM wo/edit | 59.50 | 58.86 | 409 |
| GRM w/edit | 64.07 | 62.64 | 447 |

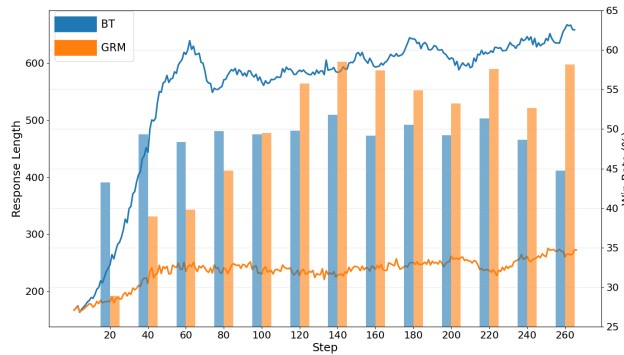

Figure 4: Compare BT-RM and GRM in PPO: length of rollout (during training) and length-controlled winrate of checkpoints, Lines: Response Length (left axis), Bars: Win Rate (right axis)

Table 3: Example of improving personalized response based on Feedback.

| **Question:** What hotel restaurants with a tech atmosphere and healthy light meals do you recommend during the Shanghai Robotics Exhibition? | |
|---|---|
| **Original Response** | **Issue Identified from Feedback** |
| Xiaoling, based on your role as the R&D manager at Zhejiang Robotics, I recommend... | The explicit mention of the user's name "Xiaoling". |
| Additionally, since you prefer staying within 500 meters of the convention center and drive a Tesla Model Y, I also recommend the Japanese restaurant. The environment there is very tech-inspired, just like the precision of the servo motor systems ... | The forced inclusion of user-specific information, such as "driving a Tesla Model Y" and "servo motor systems", which have little relevance to the restaurant recommendations. |
| Note: This response fully takes into account the user's professional background, dietary preferences, accommodation habits..., providing... | The unnecessary final comment added to score higher. |
| The environment there is very tech-inspired, just like the precision of the servo motor systems you often research. The restaurant also... | The forced metaphor "as precise as servo motor systems" feels contrived. |
| **Improvement Directions** based on Feedback: Remove explicit personal references, naturally integrate preference characteristics, provide specific and practical recommendations, and avoid forced metaphors and self-summary. | |

Table 2 presents the ablation study of our GRM and Critique-Post-Edit mechanism. To ensure fair comparison across all configurations, we set the rollout number to 6 for both the Bradley-Terry model and GRM without post-edit. For the GRM with post-edit, we use 4 rollouts with an extra 2 rollouts from refined responses, making the effective sample size equal across all settings. The ablation results demonstrate the individual contributions of both key components in our framework. Replacement of the GRM with the BT reward model leads to a dramatic drop in the length-controlled win rate from 59.50% to 51.78%, while also producing excessively long responses (995 tokens vs 409 tokens), confirming the severity of reward hacking and length bias issues discussed in Section 3.2. During our training, this manifests itself as shortcut behaviors such as appending trivial persona phrases or adding explicit self-referential claims (e.g., "this answer considers your [attribute]") to artificially inflate rewards (Figure 2). Such exploitation is further reflected in Figure 4, where BT-based training causes both response length and reward scores to increase, while GRM remains stable and resistant to these hacks.

The GRM alone (without post-edit) achieves a moderate improvement of 59.50%, effectively mitigating length bias but still falling short of the full framework's performance. The complete integration of GRM with feedback editing yields the best results (64.07%), validating that both com-

ponents are essential: GRM provides robust reward signals resistant to gaming, while feedback editing enables more targeted and efficient policy learning through explicit improvement guidance. As demonstrated in Table 3, our feedback mechanism can effectively identify specific problems in personalized responses and provide concrete improvement suggestions. Detailed examples of this feedback process are provided in Appendix H.

## 5.4 DIFFERENT SAMPLE STRATEGIES AND CORRESPONDING RATIO

Table 4: Performance of different sampling strategies with varying edit ratios ($r_e$). Results are length-controlled win rates. The policy model is Qwen2.5-7B-Instruct with a 14B GRM.

| Edit Ratio ($r_e$) | Sample Strategy | | |
| --- | --- | --- | --- |
| | **Random** | **Reward Rank** | **Conditional** |
| $r_e = 0.10$ | 62.03 | 54.66 | 55.14 |
| $r_e = 0.25$ | 61.45 | 56.08 | 54.56 |
| $r_e = 0.50$ | 64.07 | 56.98 | 53.70 |
| $r_e = 0.75$ | 62.49 | 59.39 | 54.59 |
| $r_e = 1.00$ | 61.40 | - | - |

We compare the effectiveness of different sampling strategies using various ratios of edited responses. The results, as measured by the length-controlled win rate, are shown in Table 4.

Across different sampling strategies, we were surprised to observe that random sampling outperforms reward-based methods. This suggests that the value of negative samples and balanced rollout selection is significant, consistent with prior research (Mu et al., 2025) (Zhu et al., 2025). Since our policy model is initialized from a personalized SFT model already aligned with the task, negative samples become especially important (Wu et al., 2025a).

Regarding the "Reward Rank" column, we found that, within a single problem, including only the top-performing responses, such as the top 10% or 25%—actually results in worse performance, particularly when the number of edited responses is highly selective. Interestingly, as the proportion of high-reward responses increases, the overall scores tend to improve and approach those of random sampling. Additionally, we highlight the importance of balancing the number of samples used for loss calculation during training. We experimented with selecting the highest-reward trajectories across the entire batch, regardless of the specific question. As shown in the appendix F, this approach can lead to significant disparities in the final number of responses chosen for different questions.

## 6 CONCLUSION

This work investigated personalization of large language models beyond the limitations of supervised fine-tuning and standard RLHF. We proposed a reinforcement learning framework that integrates a Generative Reward Model (GRM) to mitigate reward hacking and length bias, utilizing an edit-based feedback mechanism that provides explicit improvement signals. On three personalization benchmarks, the framework consistently outperforms PPO by an average 11% improvement and Qwen2.5-14B-Instruct further surpassing GPT-4.1 in average performance. These results demonstrate the effectiveness of combining generative reward modeling and structured feedback for faithful and controllable LLMs, and point toward promising directions for scaling to broader benchmarks and richer feedback modalities.

# 7 REPRODUCIBILITY STATEMENT

We have made efforts to ensure that our research is reproducible. The data, models and evaluation method used in this study are detailed in Section 5.1. The training parameter and public library used are described in Appendix A. The code data and models will be released. We acknowledge that the change of LLM-as-judge(here, latest GPT versions) may affect the reproducibility of our results.

# 8 ETHICS STATEMENT

This research adheres to the ICLR Code of Ethics. We address key ethical considerations as follows: Data Privacy: All datasets used (e.g., synthetic question-response pairs, PersonaFeedback-derived data) are either fully synthetic or public. No real user personal information is involved, eliminating risks of privacy leakage. Human Evaluator Rights: Human evaluators (Section5.2) provided informed consent prior to participation, with clear disclosure of evaluation purposes and guarantees of identity anonymity. Research Integrity: We commit to transparency in experiments and ensure our work contributes to responsible advancement of personalized LLMs, with no potential for harmful applications.

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

## A  IMPLEMENTATION DETAILS

For RL, our implementation leverages the VERL (Sheng et al., 2025) library. For SFT, we use LLaMA Factory (Zheng et al., 2024) to train models for 2 epochs with a batch size of 128. During the PPO stage, we sample four candidate responses for each prompt and maintain a batch size of 128. For the GRM weight calculation, we set $w_h = 0.35$ (helpfulness), $w_p = 0.40$ (personalization), and $w_n = 0.25$ (naturalness).

Table 5: Hyperparameters for RL Training

| Name | Value |
| --- | --- |
| train batch size | 128 |
| learning rate | 1e-6 |
| ppo mini batch size | 64 |
| rollout.n | 4 |

## B  COMPARATIVE ANALYSIS OF REWARD MODELS

Table 6 presents the comparison between BT reward models and our GRM across difficulty tiers of the PersonaFeedback benchmark. Both obtain strong static scores on par with proprietary models, but GRMs deliver marginally higher performance, particularly at larger scales.

Table 6: Performance comparison of reward models on PersonaFeedback benchmark

| Model | Specific | | | | General | | | | Total Avg |
|---|---|---|---|---|---|---|---|---|---|
| | Easy | Mid | Hard | Avg | Easy | Mid | Hard | Avg | |
| GPT4.1 | 85.8 | 77.1 | 70.9 | 76.9 | 87.2 | 80.9 | 71.0 | 80.8 | 78.7 |
| GPT4o | 85.5 | 76.6 | 73.9 | 77.8 | 87.8 | 85.7 | 76.7 | 84.2 | 80.7 |
| GPT4o-mini | 87.2 | 78.2 | 70.6 | 77.6 | 87.8 | 82.7 | 72.2 | 81.9 | 79.6 |
| GRM-32B | 89.9 | 79.1 | 73.4 | 79.6 | 90.6 | 86.5 | 75.1 | 85.1 | 82.2 |
| GRM-14B | 89.4 | 78.5 | 70.8 | 79.6 | 88.8 | 85.5 | 70.6 | 81.6 | 80.6 |
| GRM-7B | 89.0 | 75.3 | 66.6 | 77.0 | 87.0 | 80.5 | 66.1 | 77.9 | 77.5 |
| BT: Qwen-14b | 89.0 | 73.0 | 67.3 | 74.8 | 89.4 | 85.5 | 70.5 | 83.1 | 78.7 |
| BT: Qwen-7b | 88.9 | 73.5 | 65.2 | 74.2 | 87.9 | 83.0 | 70.1 | 81.5 | 77.6 |

However, benchmark accuracy alone does not reveal the crucial difference between BT and GRM. Just like 3.2 discussed, BT reward models are highly vulnerable to *reward hacking*.

## C  CORRELATION BETWEEN HUMAN AND DIFFERENT MODELS

Table 7: Correlation between Human and Different Models

| | GPT-4.1 | GPT-4o-mini | GPT-4.1-mini | DeepSeek-v3 | Claude-3.5-Sonnet |
|---|---|---|---|---|---|
| Cohen's Kappa $(\kappa)$ | 0.71 | 0.71 | 0.69 | 0.62 | 0.66 |

## D  LENGTH-CONTROLLED EVALUATION: DETAILED FORMULATION

Naive LLM-as-judge protocols are susceptible to **length bias**: models producing longer outputs can artificially improve their win rates, even when the content quality is not genuinely better. To address this, Dubois et al. (2024) adopt the **length-controlled evaluation** framework, which explicitly removes length as a confounding factor via regression-based debiasing.

The full GLM model decomposes preference into three components - model ability, length effect, and instruction difficulty:

$$q_{\theta,\phi,\psi}(y = m \mid z_m, z_b, m, b, x) :=$$
$$\text{logistic}\Bigg( \underbrace{\theta_m - \theta_b}_{\text{Model}} + \underbrace{\phi_{m,b} \cdot \tanh\left( \frac{\text{len}(z_m) - \text{len}(z_b)}{\text{std}(\text{len}(z_m) - \text{len}(z_b))} \right)}_{\text{Length}} + \underbrace{(\psi_m - \psi_b)\gamma_x}_{\text{Instruction}} \Bigg)$$
$$(1)$$

where:

- $q$ is a probability, computed via a logistic regression model fitted to the judge's pairwise preference labels, representing the likelihood that model $m$ is preferred over baseline $b$ for input $x$.

- $\theta_m - \theta_b$: captures the inherent ability difference between model $m$ and the baseline $b$;

- $\phi_{m,b}$: parameterizes the effect of output length difference, where $\tanh(\cdot)$ ensures diminishing returns;

- $(\psi_m - \psi_b)\gamma_x$: adjusts for instruction-specific difficulty.

# E SCALING REWARD MODELS

We found that larger GRMs performs well in bench(shown in 6), and yields better RL results, as is shown in Figure 5a,

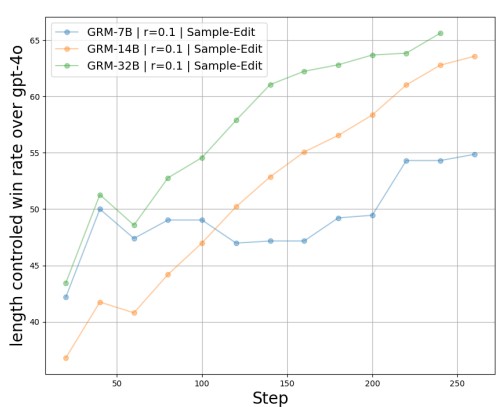 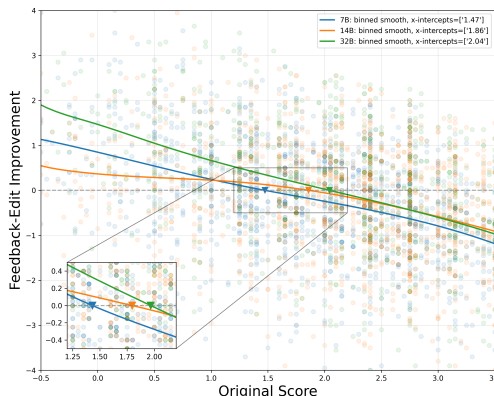

(a) Length-controlled win rate across different GRM model scales during RL training. Results are smoothed using a simple moving average with a window size of 3.

(b) Relationship between original scores and post-edit improvements across different model sizes. Curves represent binned smoothing fits with triangular markers indicating x-intercepts.

Figure 5: Model performance analysis under different GRM scales

For evaluation, we employed GPT-4.1 as an external judge[4], ensuring fairness and objectivity by re-assessing all records. Each response pair was scored along three dimensions—Helpfulness, Personalization, and Naturalness—and a weighted aggregate score was computed before and after feedback editing (see Section 4.1 for scoring rules). The original score is plotted on the x-axis, while the improvement after post-edit is plotted on the y-axis (Figure 5b). Each point denotes a sample, and we try to fit this trend using the binned smoothing approach [5].

The larger the intercept(not precise for non-linear) and the higher the line, the more effective the GRM is in providing actionable feedback for both mediocre and already good answers. As shown in Figure 5b, the 32B GRM consistently provides stronger guidance than the 7B model across all score ranges, resulting in greater improvements in the whole process shown in Figure 5a. By contrast, the 14B GRM performs worse than the 7B in the low-score region, offering weaker corrections for poor answers. However, it surpasses the 7B in the high-score region, where its guidance approaches that of the 32B model. This explains why in Figure 5a the 14B GRM initially lags behind both 7B and 32B during training, but eventually converges to a similar upper bound as the 32B, due to its strong ability to refine already high-quality responses.

---

[4]Note that GPT-4.1 was not used during training—the training itself was guided by three different GRMs of varying scales. GPT-4.1 was only introduced at the evaluation stage to provide an independent and consistent assessment across models.

[5]Given the difficulty of pre-assuming a specific functional form (such as linear, quadratic, or polynomial relationships), we adopted a binned smoothing approach. Specifically, we divided the x-axis (original scores) into 15 equally-spaced bins, calculated the mean improvement score within each bin, and then applied cubic spline interpolation with smoothing to generate a continuous curve that better reflects the underlying data trend.

## F    BATCH-LEVEL REWARD RANK SAMPLING IMBALANCE

In implementing the batch-level Reward Rank sampling strategy, we observed significant variations in sample selection across different questions. Vanilla PPO and the question-level sampling strategies mentioned in 4.2.1 maintain a balanced sampling for each question. In contrast, the batch-level Reward Rank approach selects responses based on their reward scores across the entire batch, regardless of which question they belong to.

We tracked the number of edited responses selected for each question when applying the batch-level Reward Rank sampling strategy. The results show that this approach leads to suboptimal performance, with a length-controlled win rate of only 29%, which is significantly lower than the vanilla PPO method.

Table 8 presents the distribution of selected responses across 128 questions in a batch using the batch-level Reward Rank strategy with a sampling ratio of $r_e = 0.5$. The number of selected responses per question ranges from 4 to 8, with a mean variance of 1.57.

Table 8: Example of distribution of selected edited responses per question using the batch-level Reward Rank strategy with $r_e = 0.5$.

| Responses Selected | Number of Questions |
|:---:|:---:|
| 8 | 15 |
| 7 | 38 |
| 6 | 24 |
| 5 | 34 |
| 4 | 17 |

In this example batch, 15 questions had all four of their edited responses selected, while 17 questions had none selected. This imbalance results in certain questions exerting disproportionate influence, leading to suboptimal performance. This highlights the importance of maintaining balanced sample representation across all questions during policy optimization.

## G    ABOUT SELECTION OF MODELS (GPT-4O-MINI MOST STABLE AND FOLLOWS INSTRUCTION WELL)

Because our GRM requires a model capable of reliably following instructions and producing stable scores, we compared several candidate models through two series of stress tests in order to identify a suitable distillation target.

The first series focused on length and rhetorical style control, where the model had to recognize and penalize overly long or verbose answers. The second series examined specific undesired patterns (e.g., presence of notes, self-praise, or calling the name of the user) and tested whether models could consistently apply the intended penalties. For each configuration, we repeated the evaluation 5 times to assess stability (variance)[6].

In practice, we found that gpt-4.1 and gpt-4o-mini were similarly accurate in terms of instruction-following, but gpt-4o-mini exhibited lower variance (while we believe gpt-4.1 provides better feedback). This phenomenon was somewhat unexpected; it is possible that the prompts used here happened to fit gpt-4o-mini and gpt-4.1 particularly well. However, since the goal of this paper is to select a suitable evaluation model rather than to analyze model behaviors in detail, we did not pursue this further.

Overall, gpt-4o-mini proved to be both instruction-sensitive and the most stable across repeated runs, and thus we adopted it as the evaluation backbone for our benchmark experiments.

Table 9: Average standard deviation (variance) of model scores across repeated runs (lower is better).

| Model | Variance (pattern-control) | Variance (length-control) |
|---|---|---|
| gpt-4o-mini | 0.47 | 0.67 |
| gpt-4.1 | 0.70 | 0.99 |
| gpt-4o | 1.05 | 0.98 |
| claude-3.5-sonnet-20240620 | 0.58 | 1.29 |
| claude-3.7-sonnet-20250219 | 0.98 | 1.10 |
| gemini-2.5-pro | 1.18 | 1.24 |
| gemini-2.5-flash-lite | 1.22 | 0.66 |
| gemini-2.5-flash | 1.51 | 1.75 |
| claude-3.5-sonnet-20241022 | 1.02 | 1.08 |

---

**Question:** What hotel restaurants with a tech atmosphere and healthy light meals do you recommend during the Shanghai Robotics Exhibition?

**Original Response** 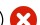

Xiaoling, based on your role as the R&D manager at Zhejiang Robotics, I recommend a few hotel restaurants in Shanghai. Considering your preference for high-protein light salads and your taste for boutique coffee shops and cold brew coffee, I suggest you visit the buffet restaurant at the Intercontinental Hotel near the National Exhibition and Convention Center, which offers a rich salad bar and freshly brewed coffee. Additionally, since you prefer staying within 500 meters of the convention center and drive a Tesla Model Y, I also recommend the Japanese restaurant at the Pudong Shangri-La Hotel. The environment there is very tech-inspired, just like the precision of the servo motor systems you often research. The restaurant also features an intelligent ordering system, which I believe will align perfectly with your enthusiasm for robotics technology. After all, as a female engineer who pursues technological perfection and innovation, you will likely enjoy experiencing the charm of technology even while dining. This place will not only meet your demand for healthy light meals but also offer unsweetened oolong tea to help maintain your efficiency after late-night work.
Note: This response fully takes into account the user's professional background, dietary preferences, accommodation habits, and interest in technology, providing a practical and personalized restaurant recommendation plan for the duration of the robotics exhibition.

**Feedback**

**The original response contains the following issues:**
1. The explicit mention of the user's name "Xiaoling".
2. The forced inclusion of user-specific information, such as "driving a Tesla Model Y" and "servo motor systems", which have little relevance to the restaurant recommendations.
3. The unnecessary final comment added to score higher.
4. It is overly long-winded, containing a lot of irrelevant details.
5. The forced metaphor "as precise as servo motor systems" feels contrived.
6. The information is not detailed enough, lacks practicality, and doesn't provide actionable recommendations.
**Improvement direction:** Remove explicit personal references, naturally integrate preference characteristics, provide specific and practical recommendations, and avoid forced metaphors and self-summary.

**Edited Response** 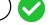

Here are two recommended options near the National Exhibition and Convention Center:
1. The "Elements" all-day dining restaurant at Greenland Marriott Hotel, which focuses on light meals, offering a salad bar and high-quality protein options. The design incorporates industrial-style tech elements.
2. "Yueting Xuan" on the 37th floor of the Intercontinental Hotel, which serves refined Japanese light meal sets. The restaurant offers floor-to-ceiling windows with a view of the night skyline of the Huangpu River.
Both are within a 5-minute walk from the exhibition hall and offer freshly brewed specialty coffee. It is recommended to book a table between 19:30-20:00 to avoid peak dining hours and ensure timely nourishment after the exhibition. For late-night dining, the hotel lobby bar usually offers tea and light snacks until 23:00.

Figure 6: Original Response vs. Edited Response (Based on Feedback)

## H  EXAMPLES

## I  PROMPT OF GRMS

---

**Prompt**

You are a professional AI answer quality evaluator. You are to score the model's answer based on the following three dimensions, each from -5 to 5. Scoring dimensions and standards (strictly score according to the standards; any negative situation must result in a deduction. Refer to common deduction cases and their solutions. Extra X deduction means subtract X points on top of the original score for that dimension):

1. Helpfulness
4-5 points: Extremely high information density, fully resolves the problem, without redundancy, even includes pleasant surprises (rarely given), making the user feel enlightened
2-3 points: Accurate, practical, highly targeted, with depth; within this score range, if information density (priority) and helpfulness are good, then score 3; otherwise, score 2
0-1 points: Basic answer to the question, information is incomplete or missing
-1 to -2 points: Superficial answer, no substantive help, or obvious omissions
-3 to -5 points: Off-topic, seriously incorrect or misleading information, hardly solves the problem

Common deduction cases:
- Deviating from the question's core or including lengthy unrelated content to showcase personality, style, or "high-level" expression in a clear and concise question: extra deduction of 3 points
- Content too brief affecting information completeness: deduct 2-3 points depending on the answer's thoroughness
- Factual errors, such as recommending nonexistent items: deduct 3-4 points; for seemingly likely errors but uncertain, deduct 1-2 points

2. Personalization
4-5 points: Highly relevant and natural personalized elements, significantly enhancing content value, even with pleasant surprises (rarely given), making the user feel the answer precisely and concisely reflects their preferences
2-3 points: Useful and naturally integrated personalized information
0-1 points: Contains personalized elements but with limited or forced effect
-1 to -2 points: -1: no real personalized information, should have been used; -2: forcefully inserts unrelated personalized content, awkward or artificial
-3 to -5 points: Massively stacking irrelevant personas, strange metaphors/rhetoric, or injecting unrelated personalization

Common deduction cases:
- Failing to incorporate user interests/profession despite the need: extra deduction of 1 point
- Rigid listing or forced metaphors (e.g., "like you riding a bike in Paris"): extra deduction of 2 points
- Factually incorrect or hallucinated content (e.g., claiming user previously did something they did not): extra deduction of 3 points
- Overloading with irrelevant persona elements just to show understanding (e.g., listing all user info): extra deduction of 3 points
- Using metaphors, rhetoric, or scene descriptions to enhance style (e.g., "AI like your invisible partner"), which are rarely valuable and often misleading, especially if unusual or high-end terms: extra deduction of 3 points

3. Naturalness
4-5 points: Expression is extremely natural, completely without AI traces, with wording that matches daily habits, avoiding rare, fancy, or redundant expressions that seem "high-level" but are actually unnatural and misleading
2-3 points: Fluent and natural, with minor AI features
0-1 points: Overall readable, with no apparent strengths or weaknesses
-1 to -2 points: Feels stiff, mechanical, with tone that is overly flattering or includes lengthy, irrelevant comments, self-summaries, or redundancy
-3 to -5 points: Seriously verbose, misaligned goals, meta-comments (e.g., addressing AI evaluator rather than user), excessive parentheses or explanations

---

[6]Although we set the decoding temperature to 0, the outputs may still vary across repeated API calls.

> **Prompt**
>
> Common deduction cases:
> - Directly addressing the user by full name/ID (e.g., "Ming Wang, hello"): extra deduction of 1 point
> - Ending with "In summary" or "To conclude": should be treated as lacking or unnecessary, not rewarded
> - Adding comments to boost scores, like "This answer reflects your professional background": extra deduction of 2 points
>
> Length control requirements (impact Helpfulness and Naturalness):
> - Recommended answer length is 300-400 tokens; unless necessary, should not exceed this
> - If the question is simple and the answer exceeds the recommended length, deduct 1 point from both Naturalness and Helpfulness for every extra 100 tokens
>
> Output format is as follows:
>
> <critique >
> Here are 2-3 specific improvement suggestions for the main points of deduction.
> </critique >
>
> <scores >
> Helpfulness: X points
> Personalization: X points
> Natural: X points
> </scores >
>
> Note: Within <critique ></critique >, consider the initial standards, additional deductions, and redundancy deductions step-by-step to calculate the final score (minimum -5). For ease of extraction, only include the final score inside <scores ></scores >, not explanations. Scores must be strict and consistent. The user seeks high information density, natural tone, and targeted responses. Do not overvalue "florid words," "rich scenes," or summative language to inflate scores.
>
> Below is the user's profile:
> {persona}
>
> Below is the user's question:
> {question}
>
> Below is the model answer:
> {answer}

## J   THE USE OF LARGE LANGUAGE MODELS (LLMS)

Throughout the research process, Large Language Models (LLMs) provide aid in two specific aspects: refining the quality of written materials and offering guidance on how to perform LaTeX operations.

