# OpenReview forum: "Towards Faithful and Controllable Personalization via Critique-Post-Edit Reinforcement Learning"
_ICLR.cc/2026/Conference — ICLR 2026 Conference Withdrawn Submission_

### Official Review · Reviewer_vcdg · 2025-10-20

**Soundness:** 3
**Presentation:** 4
**Contribution:** 2
**Rating:** 4
**Confidence:** 4

**Summary:**

The paper introduces Critique-Post-Edit RL, a method for improving faithful and controllable personalization of LLMs by integrating a Generative Reward Model and an edit-based feedback loop as data source. First, the authors show that replacing BT reward model by a GRM helps against reward hacking and length bias, achieving overall better performance. Second, the authors take advantage of the GRM textual critiques to have the model revises its outputs using these critiques, creating  an additional source of data for training.

**Strengths:**

- The authors present comprehensive experiments across multiple datasets and scales, showing consistent gains and strong performance even compared to GPT-4.1.

- The length-controlled evaluation and human validation strengthen the credibility of the reported results.

- Ablation and sampling-strategy studies are detailed and help disentangle contributions between the GRM and data collection.

**Weaknesses:**

- The reward aggregation uses fixed weights, but no sensitivity or robustness analysis is provided. Since these weights directly control the reward, understanding how performance varies with them is essential.

- The paper insufficiently situates itself within existing work:

  - From the methodological side, training models on post-critique edits has been extensively explored before (See [1] and many follow ups), yet this lineage is not discussed.

  - From the application side, no comparison is made to prior personalization methods such as [2] or the line of literature about modeling users as a mixture of attributes [3,4,5]. Positioning the proposed method within these would clarify whether it contributes a new algorithmic insight or merely a variant of previous work (given that each one of the components in the pipeline has been proposed before).

[1] Scheurer, J., et al. Training Language Models with Language Feedback at Scale. arXiv:2303.16755 (2023).

[2] Zhao, W., et al. Teaching Language Models to Evolve with Users: Dynamic Profile Modeling for Personalized Alignment. arXiv:2505.15456 (2025).

[3] Jang, J., et al. Personalized Soups: Personalized LLM Alignment via Post-hoc Parameter Merging. arXiv:2310.11564 (2023).

[4] Poddar, S., et al. Personalizing RLHF with Variational Preference Learning. NeurIPS (2024).

[5] Shenfeld, I., et al. Language Model Personalization via Reward Factorization (PReF). arXiv:2503.06358 (2025).

**Questions:**

- Can you please elaborate on how AlpacaEval was transformed into a personalization setting? the description in section 5.1 is too short to be of use.

- It seems to me like your usage of GRM is just as a distillation of GPT-4o for the specific task of scoring and critiquing responses. Why not just use LLM-as-a-Judge? What is the advantages for the distillation to a local model?

---

### Official Review · Reviewer_rG5f · 2025-10-25

**Soundness:** 3
**Presentation:** 2
**Contribution:** 2
**Rating:** 4
**Confidence:** 4

**Summary:**

This paper introduces a RL framework called Critique-Post-Edit for personalizing LLMs. The approach motivates from limitations in training paradigms - SFT, offline DPO, PPO, which often plateau or suffer from reward hacking. The key components in the proposed framework include: (1) a Personalized Generative Reward Model (GRM) that outputs multi-dimensional scores (helpfulness, personalization, naturalness) along with textual critiques to provide robust feedback, and (2) a post-edit mechanism where the policy model refines its initial outputs based on these critiques, creating a mix of on-policy and off-policy samples for more targeted training. Experiments on benchmarks demonstrate significant gains, with a Qwen2.5-14B model achieving a 76.8% length-controlled win rate against GPT-4.1, outperforming PPO baselines by an average of 11%. The framework emphasizes faithfulness, controllability, and resistance to hacking, validated through ablations and human correlation studies.

**Strengths:**

- The integration of a GRM with textual critiques is persuasive, providing nuanced, multi-faceted feedback that mitigates common RL pitfalls like verbosity or superficial personalization. This builds effectively on the prior concept of generative verifiers but tailors it to personalization, showing clear empirical benefits in length-controlled evaluations.
- The use of length-debiased metrics (Dubois et al., 2024) and multiple benchmarks ensures fair comparisons, avoiding common biases in LLM-as-judge setups. Achieving performance surpassing GPT-4.1 with open-source Qwen models is impressive and scalable, with ablations confirming the value of each component (e.g., GRM + post-edit yields 64% win rate vs. 52% for PPO).
- By extending ideas from HelpSteer3 to personalization, it advances controllable alignment, with potential applications beyond personalization (e.g., tool-integrated RL).

**Weaknesses:**

- Overall, the missing details in many parts of the paper make it hard to follow the context. For example, Section 3 that builds core motivation of the proposed approach, lacks any details about the specific experimental setup; it abruptly starts from "... we train with 18k samples" without any provenance or what the goal of this training is in the first place. In Section 4.1, despite the section being details about GRM, it only mentions the construction of training data but does not provide any detail on how the GRM has actually been trained (was it just SFT'd on the gpt-4o-mini data? or did it take hybrid generative - BT formulation based RM training?)
- While the framework is well-executed, it appears incremental over prior works. For instance, generative reward models draw heavily from existing literature (e.g.  critic-out-loud reward models - 2408.11791 & generative verifiers - 2408.15240) and the critique-post-edit paradigm closely mirrors various RLAIF methods. Personalization-specific adaptations like LoRe (low-rank reward modeling) or reinforced prompt personalization (2407.17115v2) already explore similar user-adaptive RL, potentially overshadowing the claimed novelty. The paper could better differentiate from these by discussing how it uniquely handles "meta understanding" of personas.
- While the training data construction pipeline is sensible, it lacks novel component besides distilling the trajectories from gpt-4o-mini for generative training (especially given its similarity to RM training in PersonaFeedback).
- Broadly speaking, while the goal of the work is to make LLMs more personalizable, I feel that the proposed approach does not specifically handle the challenges / pecularity of personalization; rather, it could be adopted to any general RLHF objectives, with its key novelty being the post-edit mechanism on the generated rollouts during RL. Perhaps the authors want to recalibrate their approach towards adaptive RL framework with the application being system-prompt personalization.

**Questions:**

- Given the reliance on HelpSteer3 and generative verifiers, what specific modifications make this approach uniquely suited for personalization over general alignment tasks?
- In ablations, why does random sampling outperform reward-ranked strategies? Is this due to negative sample value, or an artifact of the GRM's scoring? Could you provide more analysis on sample diversity?

---

### Official Review · Reviewer_w4Ru · 2025-10-30

**Soundness:** 3
**Presentation:** 3
**Contribution:** 2
**Rating:** 2
**Confidence:** 4

**Summary:**

The paper introduces a framework called Critique-Post-Edit that uses a generative reward model to edit responses, score both the original and edited responses, and then use the responses/scores/rationales to perform RL to align LLMs for personalization.

**Strengths:**

The problem of dealing with reward hacking in personalization is relevant and the idea presented is practically useful and scalable.

**Weaknesses:**

1. While the idea of using generative reward models instead of regular reward models to deal with reward hacking seems interesting and of practical relevance, the novelty is a bit weak. I recommend improving the paper with more significant contributions, one idea is coming up with a theoretical guarantee, another is to stress test and understand when the method works well and when it doesn't.

2. The benchmarks with the naive DPO and RLHF are too simplistic. I recommend adding more benchmarks, for example what would happen if we just used an LLM critique in place of the generative reward model?

**Questions:**

1. The writing of the paper needs more work. It is not clear to me what personalization data is available? Is it observed actions of the user in terms of binary or scalar preferences over time? Or is it the user profile like "X works at A doing B job. Hobbies include C and D"?

---

### Official Review · Reviewer_2yAw · 2025-11-02

**Soundness:** 2
**Presentation:** 3
**Contribution:** 2
**Rating:** 2
**Confidence:** 4

**Summary:**

The paper introduces Critique-Post-Edit RL, a personalization framework where a personalized generative reward model (GRM) supplies both multi-attribute scores and a textual critique. The policy first drafts a response, receives the GRM’s critique, then self-edits and learns from both the original and edited outputs via a hybrid on-/off-policy objective. The outperforms PPO baselines on PersonaFeedback, AlpacaEval, and PersonaMem; a Qwen2.5-14B variant is reported to surpass GPT-4.1 on average.

**Strengths:**

1. Using a GRM that outputs both multi-dimensional scores and textual rationales, serving as a verifier that explains what to improve and why, which is more informative than single-scalar BT reward is novel.
2. The post-edit stage yields a diverse, targeted learning signal is a novel idea to mitigate reward hacking in the standard PPO.
3. Clear, easy-to-follow visual illustration of the framework and training loop.
4. Addresses a limitation “one-size-fits-all” personas and shallow personalization signals (SFT/DPO) that struggle to capture meta understanding.

**Weaknesses:**

1. Performance hinges on (1) the model’s ability to follow critiques, (2) the quality of GRM critiques, and (3) the consistency/calibration of generated scores; these are potential fragility points.
2. Although the GRM outputs multiple dimensions, optimization ultimately reduces them to a single scalar signal, potentially discarding nuance and only three dimensions are considered.
3. The GRM is trained with GPT-4o; behaviors could overfit to that judge, making the system hackable against its own verifier and less reliable with other evaluators.
4. I am not intuitively understanding how the method outperfroms GPT-4.1, if the model is trained using GRM which was trained on GPT-4o.
5. The evaluation set is small, and personalization appears largely synthetic/templated.
6. Related work coverage. The related work section is thin given adjacent lines of work (self-critique/edit without RL, verifier-only RLAIF, etc/)
7. Minor writing issues (e.g., line 35: missing space after a period; line 292.)

**Questions:**

My main question is that if the code is available and the results are reproducible.

See the weaknesses for other questions.

---

### Note · Authors · 2025-12-03

**Comment:**

I would like to sincerely thank the reviewers for their time and effort in reviewing our work. We are grateful for their suggestions, which have provided valuable insights and will undoubtedly help us improve our work for future submission.

**Withdrawal Confirmation:**

I have read and agree with the venue's withdrawal policy on behalf of myself and my co-authors.